# Identification of BOLD engine deficiencies and suggestions for improvement based on a curated *Tachina* (Diptera) record set

**Frederik Stein**[1,2,*], **Oliver Gailing**[3]

1 Julius Kühn-Institute, Federal Research Centre for Cultivated Plants, Institute for Forest Protection, Quedlinburg, Germany, 2 Julius Kühn-Institute, Federal Research Centre for Cultivated Plants, Institute of National and International Plant Health, Braunschweig, Germany, 3 University of Göttingen, Forest Genetics and Forest Tree Breeding, Göttingen, Germany

* frederik.stein@julius-kuehn.de

## Abstract

The increasing number of Barcode of Life Database (BOLD) records per species and genus leads to contradictory species assignments within Barcode Index Numbers (BINs), serving as identifiers for the BOLD ID engine. To examine these issues, we analyzed a dataset comprising original and curated BOLD records for the genus *Tachina* (Insecta: Tachinidae), based on a previous publication. This dataset included both published and private records. We were able to assess the performance of the BOLD engine's species determination algorithm, Refined Single Linkage (RESL), and compare it to Assemble Species by Automatic Partitioning (ASAP). Additionally, we investigated the usage of BINs by the BOLD v4 ID engine. Our analysis confirmed that BOLD queries primarily rely on BINs for species identification, although some cases deviated from this pattern, resulting in species matches inconsistent with the assigned BIN species. ASAP was found to be superior to RESL due to RESL's adherence to the concept of the DNA barcoding gap. Moreover, we found that taxonomic misassignments, inconsistencies in BIN formation, and missing metadata also contribute significantly to unreliable identifications. These problems appear to stem from both algorithmic limitations and deficiencies in submission and post-submission processes. Moreover, we noted that the default mode of the BOLD v4 ID engine integrates both private and published data, leading to public records based solely on COI-based identifications. However, this issue may now be mitigated, as the BOLD v5 ID engine default mode exclusively employs published data. To enhance BOLD's reliability, we propose improvements to submission and post-submission processes. Without such amendments, the accumulation of contradictory species assignments within BINs will continue to rise and the reliability of specimen identification by BOLD will decrease.

**Data availability statement:** All data underlying the findings of this study are available in the supplementary data files provided with the following publication: Stein F, Gailing O, Moura CCM. Curating BOLD records via Bayesian phylogenetic assignments enables harmonization of regional subgeneric classifications and cryptic species detection within the genus Tachina (Diptera: Tachinidae). Annals of the Entomological Society of America. 2024:245–256. doi: 10.1093/aesa/saae018. In addition, we have included an R project as supplementary material with the current submission, which was used to generate Figs 2 and 3. This includes metadata updates not covered in Stein et al. (2024).

**Funding:** Project: Risk Management for Biotic Damage to Ensure Sustainable Forest Management (RiMa; FKZ 22019814) funded by the Federal Ministry of Food and Agriculture/FNR The funders had no role in study design, data collection and analysis, decision to publish, or preparation of the manuscript.

**Competing interests:** The funders had no role in the design of the study; in the collection, analyses, or interpretation of data; in the writing of the manuscript; or in the decision to publish the results. The authors declare no conflicts of interest.

## Introduction

The increasing record number per species and the BOLD (Barcode of Life Database) identifier Barcode Index Number (BIN) [1], driven by the growth of databases and the rising demand for DNA-based biodiversity inventories, highlights the critical need for accurate BOLD reference library entries, the largest DNA barcode database [2]. Since species identities shall uncover the ecological function of individuals, a reliable identification of the described species is essential. Especially, research in plant protection demands this reliability to reveal host-parasitoid relations [3], and to identify and classify pests [2], quarantine pests [4] or their antagonists [5].

Particularly, concerning metazoan and the commonly used mitochondrial cytochrome oxidase subunit 1 5' (COI-5P) region, challenges in species identification and efforts to curate the BOLD database have been discussed. For example, previous work [1] addressed the issue of screening in submitted sequences for *Wolbachia* spp., bacteria associated with many arthropod taxa as symbionts [6] and parasites [7]. OTU (Operational Taxonomic Unit) pipeline quality criteria, such as a minimal sequence length of 500 base pairs (bp) and less than one percent ambiguous nucleotides, have been introduced to BINs [1]. These measures ensure adequate sequence coverage [1] and consequently the discrimination of sequences based on sufficient variable sites. Furthermore, when BOLD queries do not match any sequence, the possibility that the reverse complement sequence was entered is considered. Therefore, a tool was provided to generate the forward orientation of the reverse complement sequence [8]. Recently, previous work demonstrated that Bayesian phylogenetic analysis can be employed to curate BOLD data, provided the dataset contains all contradictory species assignments [9].

However, previous work [10] criticized certain aspects of BOLD within the context of a re-analysis of an earlier study [11]. The study [11] provided the first species descriptions exclusively based on consensus sequences and BIN partitions. The primary concerns arose from the fact that BINs are generated using the "Refined Single Linkage" (RESL) algorithm, which incorporates both published and private sequences in BOLD [10]. Consequently, this raises doubts about the transparency of BINs as reliable references for specimen identification. Furthermore, the BIN formation process is not reproducible since private sequences are not accessible to users. This has led to critical views on the upcoming tendency to generate new species descriptions solely based on COI-5P sequences [11] and their automatic partitioning into species identities [10] by RESL analysis, which serve as identifiers in BOLD search queries [1]. Furthermore, the documentation of morphological specimen identification in BOLD should be improved, for example, by incorporating a mandatory reference annotation for species identification, such as species keys [12].

We employed a BOLD record set of *Tachina* Meigen, 1803 representing all contradictory species assignments within BINs up to the date of download [9]. In this study, we employed the resulting complete and curated dataset to compare the performance of the species determination algorithm RESL [1] with the more recent species determination algorithm, "Assemble Species by Automatic partitioning" (ASAP) [13].

Additionally, we tested the BOLD v4 ID engine with query sequences representing BINs and species assignments of the dataset. We aimed to evaluate the consistency of results referring to the BIN-based BOLD ID engine [1]. Additionally, we investigated whether records with species assignments based on morphological identification are consistently included in BOLD queries using the "Public Records Barcode Database". Moreover, we identified errors and gaps in BOLD metadata, such as taxonomically incorrect assignments, questionable species assignments, and questionable BIN registries. Additionally, we provide a list of possible actions and procedures to prevent the accumulation of contradictory species assignments in BINs.

We note that the current study was carried out using BOLD v4, and that BOLD v5 was launched based on BOLD v4, which was introduced in 2024. However, no publication or manual has been released yet. We would like to point out that the results page of the BOLD ID engine has been updated, but the former BOLD v4 ID engine can still be used in BOLD v5. Many processes, such as the submission process, are still based on BOLD v4, and BINs based on RESL still serve as identifiers. We also note that the BIN system is dynamic, as the BIN registry is updated monthly [14]. In addition, records assigned to *Tachina* may have been added or deleted since we conducted our analysis. However, as we consider the problems identified in BOLD to be systematic, we assume that they are still present, regardless of the recentness of the employed dataset.

## Materials and methods

### Data

For the study, we utilized BOLD data related to the genus *Tachina* with 808 BOLD records. First, the dataset, containing all existing species assignments per BIN (including all contradictory species assignments within BINs) and almost all private and published *Tachina* records, was curated based on a Bayesian phylogenetic analysis of COI-5P sequences from BOLD v4 [9]. Additionally, we modified the metadata by updating BIN assignments on September 4, 2023 (S1 Table in S1 File). To identify errors in taxonomic assignments, we generated two species assignment columns in the dataset. One column contained species names annotated in BOLD metadata, and the second one included the revised final species names based on curated records [9]. The final species assignments were based on metadata and coherence between the phylogenetic inference and the taxonomic position at the subgenus level. However, we did not revise inconsistent species assignments within subgenera because discrepancies might be due to incomplete lineage sorting, undetected species complexes and horizontal gene transfer mediated by *Wolbachia* bacteria, in addition to technical issues (e.g., misidentification, PCR contamination) [9].

According to the final species assignment, the dataset contained 21 described species, where one species was separated into two cryptic species. Furthermore, at least six additional species identities were labelled with OTU names [9]. One of these OTU names, *Tachina* nr. *magnicornis*, refers to a group of individuals considered to represent one morphologically distinguishable taxon, with the abbreviation "nr." (near) indicating uncertain but close taxonomic affinity [15]. Three of the six OTU names were allocated to species identities detected with the ASAP algorithm. For the remaining two OTU names in the BOLD metadata, we found no further information explaining why an OTU name was assigned [9]. The metadata were processed using R 4.3.0 [16] in RStudio 2023.6.2.561 [17] and the tidyverse package [18].

### Comparison of species determination algorithms

To compare the performance of the RESL and ASAP algorithms, we compared species partitions with the final species assignments [9]. Therefore, we visualized the BIN and ASAP species partitions in the final phylogenetic tree (Fig 1). Information on the allocation of BIN and ASAP partitions to records can additionally be found in S1 Table in S1 File. For the visualization, we modified collapsed sequence groups to annotate BINs and ASAP partitions. We employed the interactive Tree of Life v6 (iTOL) and the iTOL annotation editor v.1.4 [19] to modify the final tree.

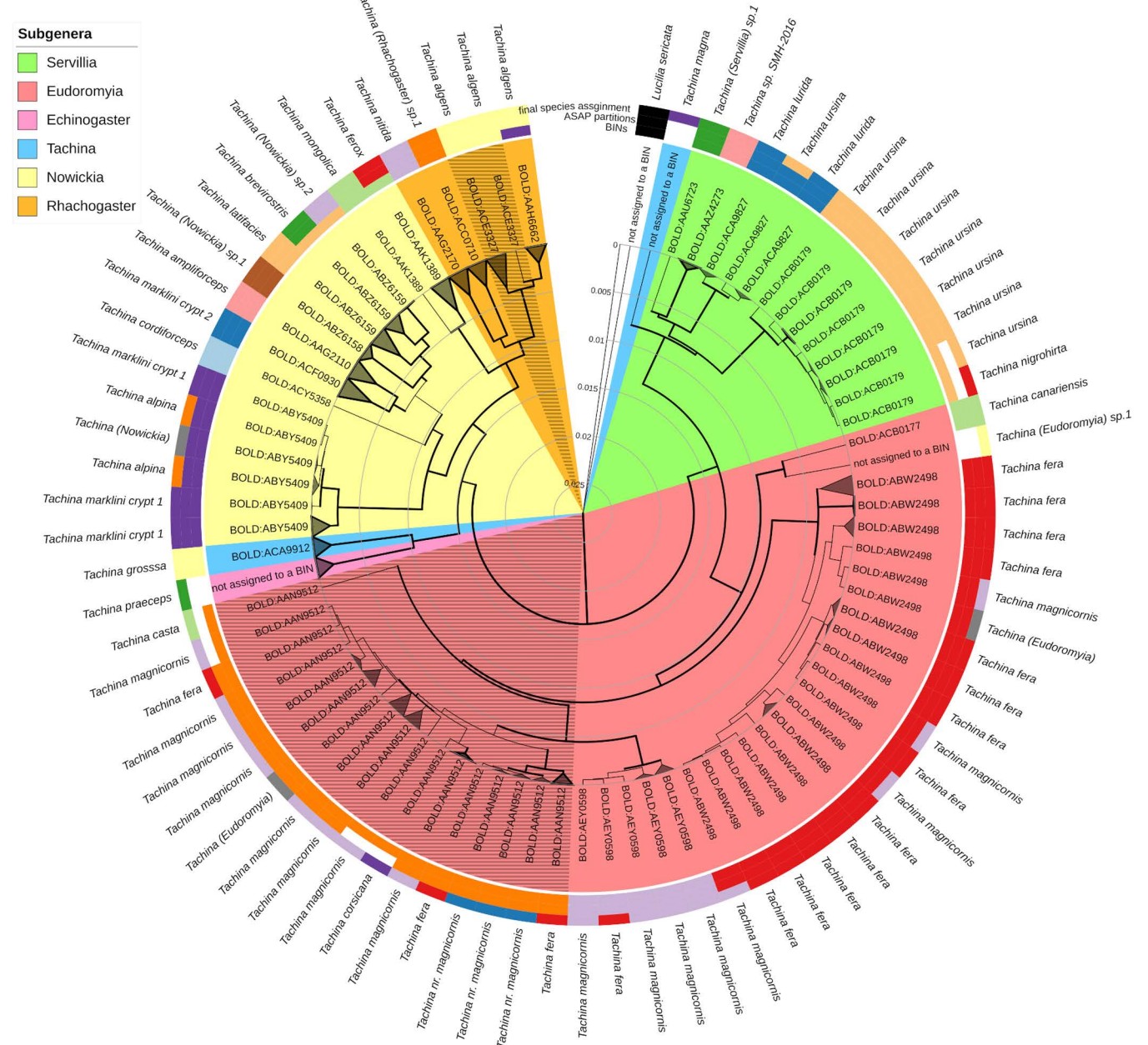

**Fig 1. Comparison of species identities detected by RESL, ASAP and clades containing final species assignments [9].** The original tree [9] was modified by adding BIN partitions and collapsing clades to the respective BIN partitions and hatching areas, indicating non-monophyletic BIN species identities formatted by the RESL algorithm. The BINs, ASAP partitions and final species assignments are differentiated within subgenera by the inner, middle and outer color strips. White stripes indicate that an assignment to a species identity was not possible. Colors were used only once within a single subgenus.

## BOLD query testing

Furthermore, we employed the record dataset to test the default mode of the BOLD v4 ID engine, to examine whether the suggested species matches on the results page corresponded to the BIN. In our experience, results appear to deviate

from this pattern in some cases. We ran default BOLD queries with a set of sequences representing different BINs and species, and also employed sequences that were assigned to species but not to BINs. For each BIN and originally annotated and later revised species name, we used the longest sequence with the lowest percentage of ambiguous nucleotides.

### Examination of publication state and annotated identification method

Moreover, the BOLD ID engine provides an option to query exclusively within the "Public Record Barcode Database," which suggests a higher degree of reliability by limiting results to published records. However, some BINs within this database, such as BOLD:AAG2170, contain only COI-based species identifications without corresponding published morphological validation. Consequently, there is a dependence on private morphological references, which is inconsistent with the expected degree of curation. Therefore, we utilized the sequence assigned to process ID "CNCTS095-18", BIN BOLD:AAG2170 and the species *Tachina nitida* (Wulp, 1882) to test the "Public Record Barcode Database" option of the BOLD ID engine. Furthermore, we visualized the distribution of annotations for identification methods (morphological, COI-based, and unknown) across the dataset according to the publication state.

### Identification and classification of errors and gaps in BOLD annotations

The documentation of errors and gaps in annotations was quantified based on counted records, which were affected by the concerned errors and gaps (Table 1). We counted records with contradictory species assignments between annotated and revised species names [9]. Moreover, we identified and categorized incorrect taxonomic assignments due to the use of outdated species names, incorrect genus assignments, incorrect species assignments, the use of subgenus names instead of genus names, and outdated COI-based identifications relying on the revised species assignments. Outdated COI-based identifications resulted from additional record uploads that caused contradictory species assignments within BINs. Furthermore, we identified records assigned to BINs but not to species, even though other records linked to the identical BIN were assigned to a single species. Since these records would have enabled a species assignment for those records lacking such assignments, we categorized this as "missing species assignments".

Furthermore, we counted records associated with "questionable species assignments". We refer here to contradictory species assignments in species clades of the final tree, which hamper unambiguous species discrimination, as it remained unclear whether inconsistencies were due to human errors or species introgression and incomplete lineage sorting [9]. Therefore, we registered records representing species assignments, which were in the minority in the concerned species clades, as "due to misidentification or species introgression". Records that we identified to produce contradictory information due to too recent speciation events were not considered, since contradictory information was presumably due to a low resolution of the COI-5P gene locus and not to human errors [9]. A second documented category was questionable species assignments "due to unknown identification method". This concerns records, which contribute to contradictory species assignments within species clades. Depending on the identification method, the concerned records had to be classified as "due to misidentification or introgression" within the same erroneous class or "due to an outdated COI-based identification" within the erroneous class "incorrect taxonomic assignment".

When sequences were assigned to a BIN that conflicted with the BOLD OTU pipeline quality criteria—which require a minimum sequence length of 500 bp and a maximum of one percent ambiguous nucleotides [1]—we noted this as a "conflicting BIN assignment" within the class "questionable BIN registry". Furthermore, we documented records, which were allocated to a BIN that incorporated at least two species conflicting with the final species assignments and ASAP partitions [9]. The third category error, "questionable BIN registry," included records with non-monophyletic species identities.

Moreover, we counted records associated with gaps and errors in the class further metadata. Further metadata was considered as metadata not directly related to taxonomic assignments or the BIN registry. Consequently, this class contained deficiencies in metadata, like missing information on geographical origin or sex. We documented records

**Table 1. Classes of errors and gaps in BOLD metadata.**

| Errors or gaps | Class | Description |
|---|---|---|
| Incorrect sequence length | Further metadata | The annotated sequence length of a record deviates from the measured sequence length. |
| Unknown identification method | Further metadata | The identification method for a record is not mentioned in the metadata. |
| No information on sex | Further metadata | The information on sex is not mentioned in the metadata of a record, despite a morphological identification. |
| No country information | Further metadata | The information on the sampling location is not given in the metadata of a record. |
| Species annotated in the wrong field | Further metadata | The species assignment of a record is registered in the field "extra info" instead of the field "identification." |
| Non-monophyletic structure | Questionable BIN registry | The species identified by BINs is non-monophyletic. |
| Entails 2 or more confirmed species | Questionable BIN registry | The BIN assigned to a record contains more than one species, which contradicts the final species assignment [9]. |
| The assignment is conflicting | Questionable BIN registry | The BIN assignment of a record contradicts the BOLD OTU pipeline criteria [1]. |
| Due to an unknown identification method | Questionable species assignments | The species assignment of a record contradicts the species assignments predominant in a species clade. If the identification was COI-based, this information should be removed. |
| Due to misidentification or species introgression | Questionable species assignments | The species assignment stands in contradiction to the species assignments predominant in a clade. If the identification was due to human failures, it had to be removed. |
| Missing species assignment | Incorrect taxonomic assignment | The record is assigned to a BIN, but not to a species, although a COI-based identification is possible. |
| Outdated COI-based identification | Incorrect taxonomic assignment | There is a species assignment of a record relying on a COI-based identification, although there are contradictory species assignments in the assigned BIN |
| The genus name is a subgenus | Incorrect taxonomic assignment | The subgenus name was applied as the genus name of a record. |
| Outdated species name | Incorrect taxonomic assignment | The assigned species name of a record is outdated. |
| Incorrect genus | Incorrect taxonomic assignment | The record was wrongly assigned to the genus *Tachina*. |
| Incorrect species | Incorrect taxonomic assignment | The record was assigned to the wrong *Tachina* species. |

where, despite morphological identification, information on sex was missing, as sex determination is part of morphological identification in Tachinidae. Moreover, morphological identification of female Tachinidae is challenging due to fewer distinct morphological characters available to identify the species [20,21]. Consequently, the information could have contributed to assessing the reliability of morphological identification [9]. We also counted records where the identification method was not stated. Moreover, we reported incorrect annotated sequence lengths. To this end, we measured sequence length and the number of ambiguous nucleotides with BioEdit Vers. 7.2.5 [22] in the fasta file employed. Subsequently, we compared the annotated sequence length with the measured sequence length. However, we reduced the annotated sequence length by one bp, in case the length exceeded 657 bp, since the employed fasta file used sequences with a maximum length of 657 bp due to the adjustment to the reading frame [9]. Furthermore, we registered records without a sampling location registry that conflicted with the metadata requirements for submission [23]. We excluded the sequence with the process ID "AFTAC096-20" for further metadata analysis, because we received the sequence via email and had no direct access to metadata on BOLD [9]. If the species assignment was registered in the "extra info" field and not in the "identification" field, the records were documented as "species annotated in the wrong field", as this blocks the sample identification by the BOLD query if these assignments represent the only representatives of their species.

## Results

### Accordance and discrepancies between final species assignments, BINs and ASAP partitions

For ten species identities, involving five described species, one cryptic species, and four OTU species assignments, we found complete agreement between BIN, ASAP partitions, and final species assignments, forming monophyletic clades. In one instance, BIN BOLD:ACB0179 did not align with the final species assignment. RESL grouped *Tachina ursina* (Meigen, 1824), sequences and a *Tachina nigrohirta* (Stein, 1924) sequence into the same BIN. However, the *Tachina nigrohirta* sequence was too short (<500 bp) to be considered by ASAP. Consequently, ASAP did not evaluate this sequence; however, RESL still assigned it to BOLD:ACB0179, which contradicts the OTU pipeline criteria. The final species assignment, however, recognized *Tachina nigrohirta* as distinct from *Tachina ursina*, as *Wolbachia*-mediated species introgression or incomplete lineage sorting of closely related species could not be excluded [9]. For three single-sequence clades representing two described species, *Tachina magna* (Giglio-Tos, 1890), *Tachina praeceps* Meigen, 1824 and one sequence assigned to the OTU name *Tachina* (Eudoromyia) sp.1, species assignments could not be confirmed by RESL and ASAP algorithms, because the sequences were too short.

Monophyletic clades and the ASAP partition confirmed six species identities; however, the BINs did not match these species identities. Specifically, the RESL algorithm was incapable to separate monophyletic branches representing *Tachina mongolica* (Zimin, 1935) and *Tachina ferox* Panzer, 1806 & Panzer, grouping them into a single clade corresponding to BIN BOLD:AAK1389 (Fig 1). Likewise, the species identities of closely related taxa *Tachina latifacies* (Tothill, 1924), *Tachina brevirostris* (Tothill, 1924) and *Tachina* (*Nowickia*) sp.2 were assigned to a single BIN, BIN BOLD:ABZ6159. For *Tachina algens* Wiedemann, 1830, two BINs (BOLD:ACE3327; BOLD:AAH6662) suggested putative cryptic species, but ASAP and final species assignments indicated only one species. The BIN BOLD:ACE3327 represented a paraphyletic group since it had a common ancestor, but did not include BIN BOLD:AAF6662 (Fig 1).

In five cases, species identities indicated by ASAP partitions and BINs were consistent. However, the final species assignments differed because revisions based on the phylogenetic relationships of closely related species were not performed [9]. This decision was made due to potential species introgression or incomplete lineage sorting among these species. Notably, within the subgenus *Eudoromyia* Bezzi, 1906, *Tachina fera* (Linnaeus, 1761) and *Tachina magnicornis* (Zetterstedt, 1844) were assigned to three putative species identities as detected by ASAP partitions and BINs. However, BIN BOLD:AAN9512 and the corresponding ASAP partition differed by one record, as ASAP ignored a sequence shorter than 500 bp, while RESL incorrectly included it. Therefore, the BIN included the record of *Tachina casta* (Rondani, 1859), rendering the putative species identity represented by the BIN polyphyletic.

Additionally, the monophyletic clade primarily containing *Tachina magnicornis* species assignments encompassed two ASAP partitions and the corresponding BINs BOLD:AAN9512 and BOLD:AEY0598 and included the three species assignments, *Tachina fera*, *Tachina* nr. *magnicornis* and *Tachina corsicana* (Villeneuve, 1931). However, sequences allocated to two of the three species assignments were too short to be taken into account by ASAP and were assigned to BINs, contradicting BOLD OTU pipeline quality criteria. The third OTU species *Tachina* nr. *magnicornis*, was indistinguishable from *Tachina magnicornis* and *Tachina fera*, likely due to incomplete lineage sorting. Similarly, the RESL and ASAP algorithms failed to distinguish between the clade assigned to *Tachina marklini crypt 1* and *Tachina alpina* (Zetterstedt, 1849) (Fig 1). Furthermore, the clade is predominantly composed of *Tachina lurida* (Fabricius, 1781) sequences, and additionally contains one sequence assigned to *Tachina ursina*.

### BOLD query results

In most cases, the results of a default BOLD v4 query corresponded to the BINs assigned to query sequences. However, in some cases, no match to the species could be achieved. This was the case when the query sequences were assigned to a BIN without a species assignment, when the species name was annotated in the wrong field or when records were assigned to a species and but not to a BIN (S2 Table in S1 File).

Moreover, in some cases, the results did not correspond to species assignments allocated to the BIN of the query sequence (Table 2). For *Tachina marklini* (Zetterstedt, 1838) sequences assigned to the BIN BOLD:ACF0930, which only incorporates species assignments of *Tachina marklini*, the species *Tachina ampliforceps* (Rowe, 1931) was also suggested in the results. Vice versa, for sequences allocated to the species name *Tachina ampliforceps* and assigned to BIN BOLD:AAG2110, additionally *Tachina marklini* was suggested as a possible species match (Table 2).

We identified a further discrepancy between the BOLD query results and the assigned BIN of the query sequences for sequences assigned to BIN BOLD:AEY0598. This BIN comprised the species names *Tachina magnicornis*, *Tachina fera* and the outdated species name [24] *Tachina nupta* (Rondani, 1859). However, the query results yielded the species names *Tachina magnicornis*, *Tachina fera* and *Tachina* nr. *magnicornis*. Hence, the record assigned to *Tachina nupta* was disregarded. *Tachina* nr. *magnicornis* was exclusively allocated to BIN BOLD:AAN9512, while *Tachina magnicornis* and *Tachina fera* were allocated to BIN BOLD:AEY0598 and BIN BOLD:AAN9512. Therefore, the BOLD query with sequences assigned to BIN BOLD:AEY0598 triggered partial species assignments of BIN BOLD:AAN9512 (Table 2).

For sequences assigned to BIN BOLD:AAN9512, we also found differences between query results and species assignments of the BIN. For sequences assigned to *Tachina magnicornis*, *Tachina corsicana*, *Tachina fera* and *Tachina* nr. *magnicornis*, matches to *Tachina magnicornis*, *Tachina fera*, *Tachina* nr. *magnicornis* were found, while matches to *Tachina corsicana* and *Tachina casta*, not corresponding to BOLD OTU pipeline criteria (sequence length < 500 bp), were not detected (Table 2). Moreover, the BOLD query with the sequence assigned to *Tachina casta* did result in no match. Merely, sequences assigned to the species *Tachina fera* were reported as the nearest reference sequence. For BIN BOLD:ACB0179, which is assigned to the species *Tachina ursina* and *Tachina nigrohirta*, query sequences of both species yielded only *Tachina ursina* as a species match and ignored the record assigned to *Tachina nigrohirta*, not meeting BOLD OTU pipeline criteria (Table 2). Moreover, the BOLD query with the "Public Record Barcode Database" as reference library resulted in a match with the species *Tachina nitida* assigned to BIN BOLD:AAG2170.

## Publication state

For 22 annotated species names and 376 published records, only three records (0.8% of all published records with species assignment) were available with a documented morphological identification in metadata. Nonetheless, based on personal communication [9], additional 12.8% of samples were identified morphologically in published records. In contrast, for private records, over 55% of species assignments relied on morphological identifications, and 75% when considering information received via email (personal communication). Morphological identifications were also conducted for more than half of the entries published after our request. Considering all sequences with species assignment, more than a quarter of all records were identified morphologically, 61.7% were identified solely based on COI-5P sequences, and for 12.3% the identification method was unknown (Fig 2).

## Errors and lacks in BOLD annotations

Based on metadata [9], we detected incorrect taxonomic assignments. Nine records were wrongly assigned to the genus *Tachina* or to the incorrect species. For 17 sequences the subgenus name *Nowickia* Wachtl, 1894 was annotated as a genus name, and six sequences were assigned to junior synonyms. Additionally, for 30 sequences, the COI-based identification was outdated, and the taxonomic assignment should be removed. Conversely, species assignments for 133 sequences were missing, although a COI-based identification would have been possible using the dataset. However, for 32 of 133 records, the species assignment was only possible based on the private annotations combined with ASAP partitions.

Moreover, we found records with "questionable species assignments". We identified 7 records in the data with species assignments that were inconsistent with the majority of species assignments within the species clades. Furthermore, there

**Table 2. Selection of BOLD query results deviating from reference BINs. A complete overview of BOLD queries can be found in the supplement (S2 Table in S1 File).**

| Annotated species | | Tachina marklini | Tachina ampliforceps | Tachina magnicornis | Tachina magnicornis | Tachina casta | Tachina corsicana | Tachina nigrohirta | Tachina ursina |
|---|---|---|---|---|---|---|---|---|---|
| Process ID of query sequences | | CNCDT783–12 | BBDCP692–10 | NLDIP289–12 | BGTAC045–10 | GBDP9352–10 | GBDP9354–10 | GBDP9356–10 | TACFI100–12 |
| BOLD OTU pipeline quality criteria | sequence length > 500 bp | fulfilled | fulfilled | fulfilled | fulfilled | not fulfilled | not fulfilled | not fulfilled | fulfilled |
| | > 1% ambiguous nucleotides | fulfilled | fulfilled | fulfilled | fulfilled | fulfilled | fulfilled | fulfilled | fulfilled |
| Assigned BIN | | BOLD:ACF0930 | BOLD:AAG2110 | BOLD:AEY0598 | BOLD:AAN9512 | BOLD:AAN9512 | BOLD:AAN9512 | BOLD:ACB0179 | BOLD:ACB0179 |
| Species assigned to BIN | | Tachina marklini | Tachina ampliforceps | Tachina magnicornis<br>Tachina fera<br>Tachina nupta | Tachina magnicornis<br>Tachina fera<br>Tachina nr. magnicornis<br>Tachina corsicana<br>Tachina casta | Tachina magnicornis<br>Tachina fera<br>Tachina nr. magnicornis<br>Tachina corsicana<br>Tachina casta | Tachina magnicornis<br>Tachina fera<br>Tachina nr. magnicornis<br>Tachina corsicana<br>Tachina casta | Tachina ursina<br>Tachina nigrohirta | Tachina ursina<br>Tachina nigrohirta |
| Species matched in BOLD query | | Tachina marklini | Tachina ampliforceps<br>Tachina marklini | Tachina magnicornis<br>Tachina fera<br>Tachina nr. magnicornis | Tachina magnicornis<br>Tachina fera<br>Tachina nr. magnicornis<br>Tachina corsicana<br>Tachina casta | No species match | Tachina magnicornis<br>Tachina fera<br>Tachina nr. magnicornis | Tachina ursina | Tachina ursina |
| Discrepancy between BOLD query results and assigned BIN | Missing | | | Tachina nupta | | All species | Tachina corsicana<br>Tachina casta | Tachina nigrohirta | Tachina nigrohirta |
| | Additional | Tachina ampliforceps | Tachina marklini | Tachina nr. magnicornis | | | Tachina casta | | |

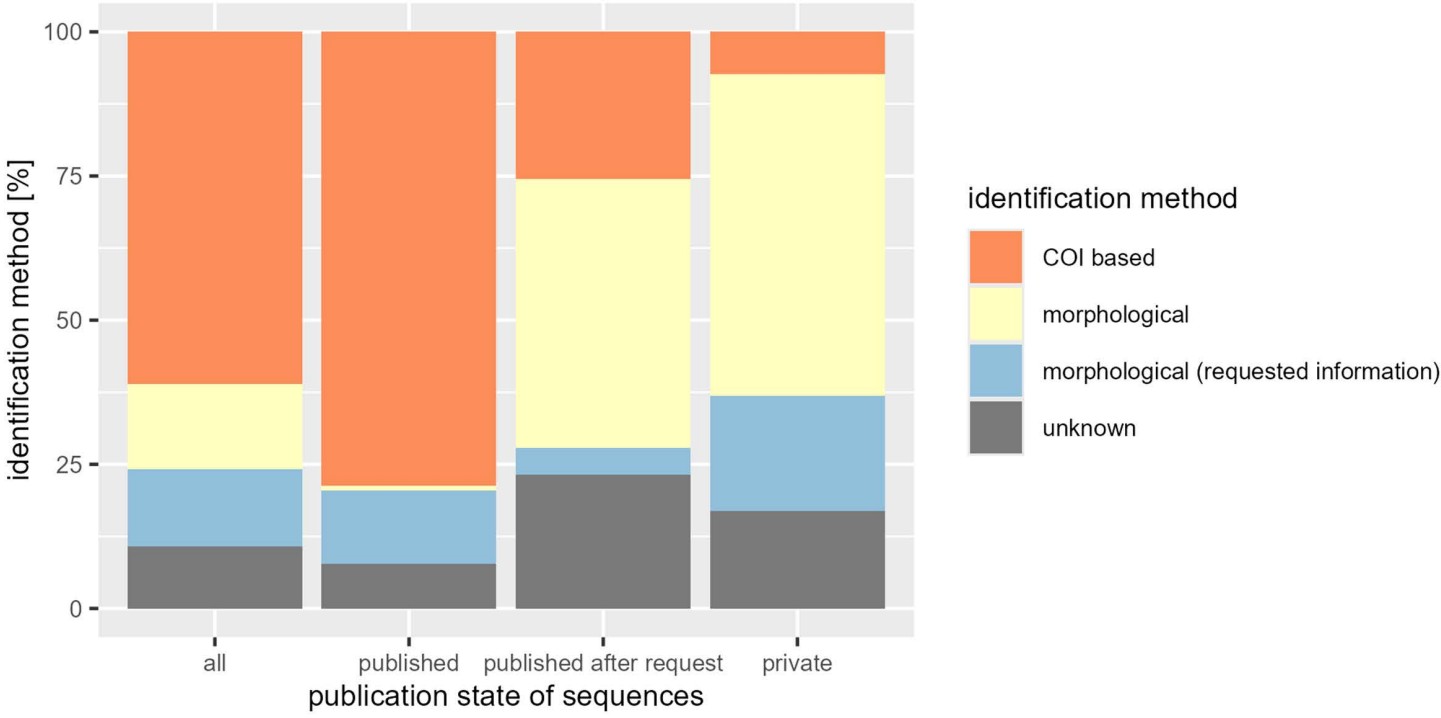

**Fig 2. Publication state and identification method of sequences assigned to species.**

were 32 records where the identification method was unknown and the species assignment would have to be removed, in case the identification method was COI-based, as they were allocated to BINs with conflicting species assignments based on morphological identifications.

Beyond that, we spotted questionable BIN registries. We detected 19 sequences with BIN assignments conflicting with OTU pipeline criteria [1], because sequences were shorter than 500 bp or the frequency of ambiguous nucleotides exceeded 1%. Additionally, we registered 105 records in BINs, each of which included at least two described species or species identities. Additionally, we counted 230 records assigned to non-monophyletic putative cryptic species identities represented by the BINs BOLD:ACE3327, comprising a portion of *Tachina algens* records and BOLD:AAN9512, including records assigned to *Tachina magnicornis*, *Tachina fera*, *Tachina corsicana* and *Tachina casta* (Fig 1).

Furthermore, we detected false registrations and gaps in further metadata. For 342 records, the annotated sequence length was 1–377 bp longer than the measured sequence length, for 66 records, the sex was not specified, although a morphological identification was conducted (Fig 3), and for 90 records, the identification method was not documented in metadata. Furthermore, the country was not specified for 18 records, which is contrary to the submission requirements. According to annotations, these records were mined from GenBank. Furthermore, for 14 records, the species name was annotated wrongly in the field "extra info" instead of the field "identification" (Fig 3). This applied to all records to which the species *Tachina latifacies* and *Tachina brevirostris* were assigned in metadata.

## Discussion

Our analyses revealed weaknesses in the employed species determination algorithm, RESL, as well as in the BOLD ID engine, BIN registry, data submission, and post-submission processes, which promoted errors and gaps in metadata. Notably, communication about how private sequences influence public data remains insufficient. While our analysis

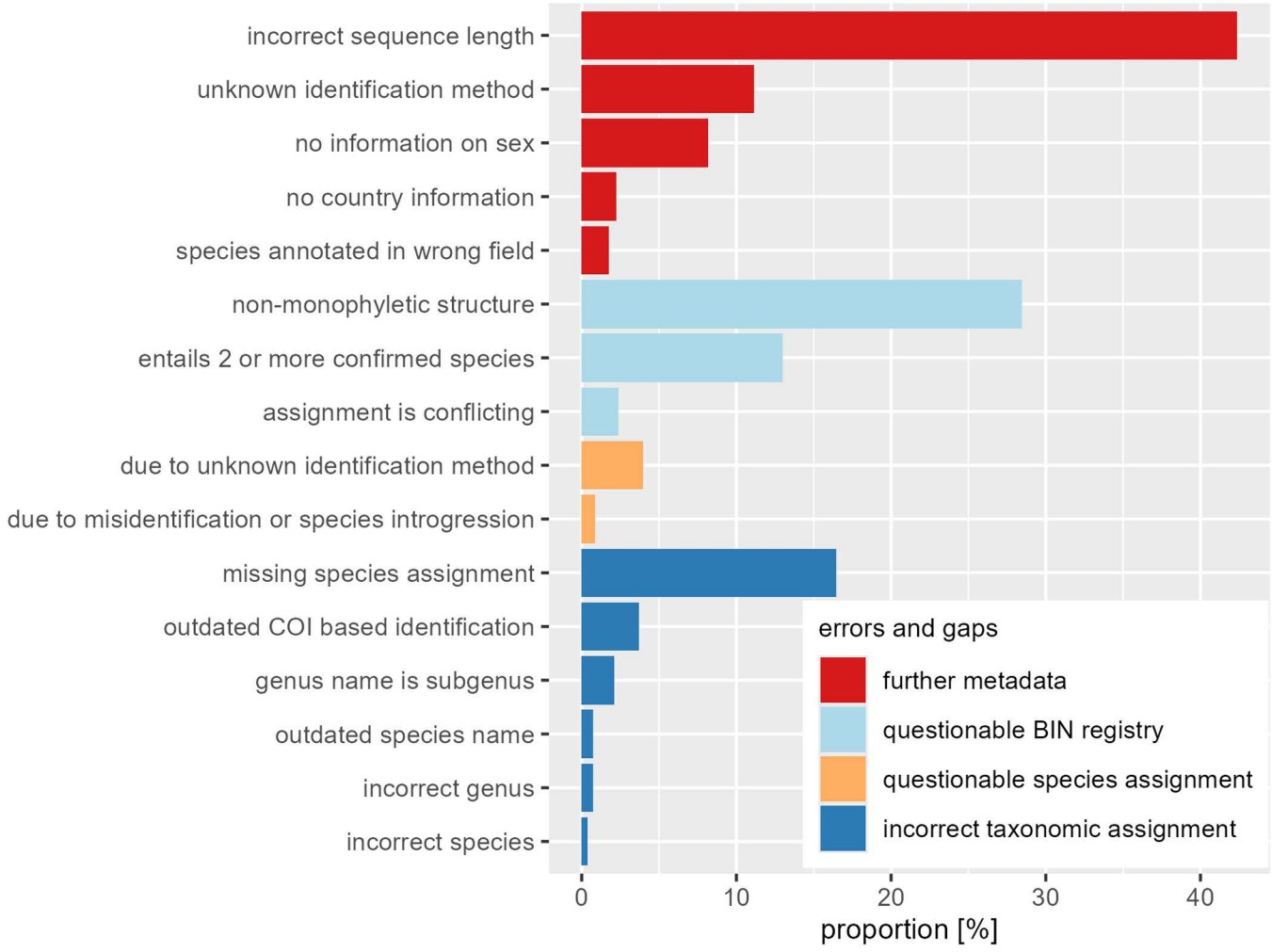

**Fig 3. Proportion of errors and gaps in BOLD annotations of *Tachina* records (n = 812) obtained from BOLD.**

focused on a representative dataset from the genus *Tachina*, we believe that the underlying mechanisms we identified—such as how metadata inconsistencies and algorithmic limitations can affect BIN assignments—are relevant across the entire BOLD system. We acknowledge, however, that further large-scale analyses across diverse taxa would be necessary to generalize the extent and quantitative impact of these issues across all BOLD records.

## ASAP yields more reliable species assignments than RESL

Both the RESL [1] and ASAP algorithms [13] yielded some results partly inconsistent with the final species assignments [9]. However, ASAP was superior in species determination. For example, it separated the group representing BIN BOLD:ABZ6159 into three species identities, including *Tachina latifacies*, which lacked a clear barcoding gap. Similarly, it split BIN BOLD:AAK1389 into two species, despite *Tachina ferox* having a minimal barcoding gap of 0.01% genetic divergence [9]. Consequently, the species *Tachina latifacies* and *Tachina ferox* represent cases where lineage sorting is completed but a barcoding gap has not yet formed.

Furthermore, unlike RESL, ASAP did not separate *Tachina algens* records into two partitions; instead, it grouped all *Tachina algens* records correctly into a single partition. *Tachina algens* might represent an evolutionarily old species, where accumulated mutations over time resulted in high intraspecific divergence, especially in cases of secondary contact between phylogeographical lineages [25]. In such situations, the genetic distance between some genotypes within the species can exceed the average distance to other closely related species, resulting in a species without a barcoding gap. ASAP's strength lies in its dual-metric approach: it evaluates both barcoding gap probabilities and hierarchical clustering, combining them into an ASAP score. The partition with the lowest score is considered the most likely [13]. Therefore, ASAP is capable of neglecting the concept of the DNA barcoding gap where necessary.

One exception was the partitioning of *Tachina magnicornis* into BINs BOLD:AAN9512 and BOLD:AEY0598 by both RESL and ASAP, which contradicted final assignments [9]. The observed polyphyly in BIN BOLD:AAN9512 was caused by a *Tachina casta* sequence shorter than 500 bp, which violated BOLD OTU pipeline criteria [1]. ASAP excluded this sequence from partitioning, resulting in a monophyletic ASAP partition. This raises the unresolved question of whether ASAP would have also produced a polyphyletic partition if the *Tachina casta* sequence had met the length threshold. Other discrepancies between algorithm outputs and final species assignments likely stem from insufficient gene locus resolution [9].

Among the affected BINs, our dataset covered over 85% of records, including all conflicting cases. Thus, it is unlikely that adding more sequences would alter ASAP's partitioning. Moreover, the concordance between ASAP and final species designations is not circular: most species were pre-identified in metadata and independently confirmed by ASAP, with the only exception of *Tachina* (*Nowickia*) sp.2 [9].

Therefore, we conclude that where the gene locus itself did not limit COI-based separation, the ASAP algorithm outperforms RESL in *Tachina* species determination. Although broader comparisons are limited by access to private BOLD data [9], we expect ASAP's improved performance to extend to other arthropod groups where high intraspecific divergence overlaps with interspecific distances (e.g., Diptera [25], hemipteran Miridae [26], Collembola, or Oribatida [27]). Given that ASAP explicitly incorporates the barcoding gap along with additional parameters, we expect it to perform at least as effectively as RESL in arthropod groups with distinct barcoding gaps.

Given ASAP's ability to incorporate additional parameters beyond the DNA barcoding gap, we recommend replacing RESL with ASAP (or similar algorithms) for generating BINs in BOLD. Earlier studies anticipated the development of improved species delimitation tools [1]. While employing additional clustering algorithms may be beneficial [10], this would also increase computational demands and could exceed the performance capacity of BOLD's servers.

## Amendment of the results page

Species suggested by the BOLD v4 ID engine sometimes deviated from reference BINs. Using the "Public Record Barcode Database", the BOLD v4 ID engine listed species matches even when no supporting records with morphological identification were available. Additionally, records assigned to BINs without species assignments yielded no results with the BOLD v4 ID engine.

When query sequences returned species names that conflicted with the BIN allocated to the query sequence, we surmised that the BIN assignment history played a role. This was confirmed by the delta view, which indicated that divergent species assignments came from BINs the query sequences were previously assigned to. Such inconsistencies likely result from a temporal lag between downloadable metadata and the metadata used by the BOLD v4 ID engine. The engine's desirable exclusion of records that violate BOLD OTU pipeline criteria [1] may also reflect this temporal lag. If such a temporal gap exists, we recommend closing it.

Species listed triggered by the BOLD query using the "Public Records Barcode Database" confirmed that morphological identifications are not required, as no such records were present in the reference library. Consequently, the species assigned to the record that allowed the triggering of BIN BOLD:AAG2170 was identified based on a reference sequence

that was once assigned COI-based. Therefore, the reference sequence for this COI-based identification must have relied on private data in the absence of published alternatives, suggesting that the higher degree of data curation of such a BOLD query is not guaranteed. However, a BOLD query using the 'Public Record Barcode Database' likely yielded no species match when based solely on morphological references, as only three published records had been morphologically identified as *Tachina* at the time we downloaded the sequences.

Another issue is that BINs lacking species-level assignments yield no matches when sequences assigned to these are entered into the BOLD v4 ID engine, suggesting a lack of reference records. Furthermore, matched BINs are linked on the result's webpage only if there is a single species match; otherwise, potential species matches are listed without naming the matched BIN.

Consequently, we advocate for a more transparent and comprehensive presentation of results. We recommend listing the matched BIN, all species assigned to it, and any additional BINs containing species assignments of the matched BIN. Additionally, we suggest presenting species names included in one BIN and sharing the same epithet in one line indicating synonymy. It is also useful to indicate the number of morphologically determined specimens for each listed species name. Moreover, we recommend that the neighbor-joining tree function annotate each BIN and include sequences from all species within the same genus or subgenus as the matched BIN.

The realization of these recommendations would provide an initial overview of tentative cryptic species and the distribution of conflicting species assignments across BINs. Thus, even if species introgression exists, this would allow users to gauge the probability that their sequences correspond to one of the conflicting species assignments. Ideally, BINs without species assignments should also be displayed so that users are informed when a BIN is matched, even if no species match is available.

The new BOLD v5 ID engine partially addresses the reported issues. The public database is now the default mode, and a species match based on the BIN appears on the results page only if the matched BIN contains one species assignment; otherwise, solely the genus name is shown. Moreover, the Process ID and BIN have been added to the closest records list, allowing users to trace BIN matches without species assignments and view species assigned to the matched BIN. Unfortunately, the neighbor-joining tree function has been removed. Likewise, for the BOLD v5 ID engine, the "Animal Library (Public + Private)" mode—previously the default mode of the BOLD v4 ID engine—no longer displays the publication state of the closest records. Furthermore, we note that the BOLD v5 ID engine automatically adjusts the orientation of sequences entered in reverse, but does not display that the sequence has a reverse orientation.

### Sequence submission process optimization by focusing on the identification method and introducing a third data state

We consider several errors and annotation gaps to stem from the submission process. In particular, incorrect species assignments are at least partially due to poor communication by BOLD about published and private sequences. Although the BOLD v4 manual states that private entries are also included in the results of default BOLD database queries [28], this is not communicated in the chapter on the submission process or during the submission process. The higher number of private records with morphological IDs suggests users hesitate to publish uncertain morphological identifications. Consequently, uncertain morphological records affect BOLD v4's default mode, which included both private and public data. As the BOLD v5 default mode uses only public data, the issue is expected to be mitigated, though COI-based identification via other query modes remains possible (Table 3).

Nevertheless, we suggest informing users during record submission that private records can be employed by the BOLD v5 ID engine via two search options (Animal-species-level library, public + private; Animal library, public + private). Furthermore, we recommend introducing a third data state—preliminary submitted records—alongside private and published records. These would be excluded from BIN formation by the employed species determination algorithm (Table 3), but an automated BOLD query could still provide possible species and BIN matches, as well as the record number per species associated with morphological identification for the submitted sequences.

**Table 3. Recommendations to amend the BOLD engine. Recommendations in bold text are considered in BOLD v5.**

| Issues of BOLD v4 engine | Recommendation | Solution on BOLD v5 |
|---|---|---|
| RESL algorithm underperforms compared to ASAP due to reliance solely on the barcoding gap | Employ a more recent algorithm (e.g., ASAP) that uses at least one additional metric besides the barcoding gap. | No |
| Species assignments for one BIN based on public data often lack morphological references | **List all species assigned to the matched BIN**, and display the number of morphologically identified records for each species assignment within the BIN as BOLD query results. | Partial |
| BINs without species assignments cannot be matched | **Display the matched BIN even without species assignments as BOLD query results.** | Yes |
| The default BOLD query mode is based on private and public references | **Change the default mode to 'Public References Only'.** | Yes |
| Lack of communication that private data can affect public references | Improve transparency during data submission and clearly communicate the impact of private records on public records. Introduce a third data state ("preliminary") and set the **default mode to public references only.** | Partial |
| Missing metadata on identification method, identification references, authority of first description, sex and subgenera | Revise submission datasheets: introduce mandatory, default options for identification method; mandatory field registrations for authority of first description, introduce a subgenus field, and encourage users to give information on sex where applicable. | No |
| No clear guidance on the use of OTU names | Prohibit the use of OTU names, **as BINs without species assignments can now be matched.** | Partial |
| Metadata of mined records from GenBank often has incorrect sequence length | Apply automated sequence length validation retroactively to older records, particularly those from GenBank imports. | No |
| Record history traceability requires manual inspection of the delta view for public data | Automate record history in metadata (e.g., submission data, BIN assignment) and introduce a BIN-level delta view. | No |
| Records in BINs without species assignments and no conflicting species assignments | Automatically assign species names if BINs contain no conflicting species assignments. | No |
| Records in BINs with conflicting species assignments | Contact the responsible data manager for verification of species assignments; implement an automated bioinformatics protocol to assess BIN phylogeny. | No |
| COI-based identifications are causing conflicting species assignments within a BIN | Remove COI-based species names from BINs where conflicts cannot be resolved. | No |
| Records assigned to BINs that do not meet OTU pipeline quality criteria | Enforce compliance with OTU pipeline requirements (e.g., sequence length, sampling location). | No |
| Species matches do not correspond to the current BIN or metadata | Investigate mismatches between BOLD ID engine outputs and downloadable metadata; improve synchronization between BOLD ID engine and downloadable metadata. | Unclear |

Moreover, we recommend amending data sheets for the record submission by creating a separate section addressing the often-neglected issue of identification, allowing assessment of the species assignment from a third person's perspective. Furthermore, we argue for mandatory documentation of the identification method. Additionally, registering sex information during submission should be encouraged, especially for adult specimens in arthropod groups where species identification requires predominantly sex determination, such as several dipteran families (e.g., Tachinidae, Sarcophagidae [20,29]) and hymenopteran Symphyta, where the sex is often obvious or easily determined (Table 3) [30].

Furthermore, we consider providing default options for the identification method field such as "COI-based", "morphological", and "others" along with a second optional field to specify the chosen identification method. We support previous recommendations [12] to add two mandatory annotation fields: the identification reference (such as species keys) and the author of the original species description, in accordance with the International Code of Zoological Nomenclature [31]. These measures would help avoid synonyms, facilitate species identification in BOLD, and enable tracing of

morphological identifications. If morphological identification is uncertain, an extra field should document possible species without assigning an uncertain species name in the "identification" field.

Moreover, we propose an optional subgenus field in the taxonomy section, along with a mandatory field for the author of the first subgenus description, for cases where subgenus assignment is made. This provides detailed information about the taxonomic position, to specify uncertain morphological identifications and detect incorrect species assignments within genera [9]. Another taxonomy-related issue is the unclear use of OTU names in metadata, likely assigned to aid identification via BOLD queries. However, since the new BOLD v5 ID engine displays BINs without species assignments, OTU names employment could be reduced prospectively. Regarding database errors from species assignments in the non-prescribed "extra info" field, current submission rules appear adequate if strictly enforced (Table 3).

## Automatic generation of metadata and integration of a Bayesian tool evaluating questionable species assignments post-submission

Besides submitter-created metadata during the submission process, some—like BIN assignment and sequence length—is generated automatically. However, this has not always been consistent, as some annotated sequence lengths are incorrect, likely due to older sequence uploads. Especially for sequences mined from GenBank, additional checks for sequence length seem to be missing. Furthermore, we recommend adding automatically generated fields for the dates of sequence upload, species assignment, and BIN assignment to improve traceability of record history. Currently, these are exclusively accessible in the delta view of single published records and are laborious to obtain for multiple records [10]. Therefore, developing a delta view for BINs would enable rapid tracking of BIN formation and species assignment regardless of publication status (Table 3).

The missing species assignments may occur because the BINs assigned to new sequences at upload were not yet linked to species. We recommend automatic species assignment when the BIN does not contain contradictory species assignments. For BINs without species assignments in BOLD metadata, we recommend an automated message via the workbench to data managers of the respective sequences, requesting possible morphological sample identification (Table 3). However, responsible data managers should be contacted only once per specimen to avoid pressuring them, in cases morphological identification is uncertain.

If a new record's species assignment causes contradictory BIN information, action is required by BOLD. We recommend verifying whether the new sequence disrupts monophyly among morphologically identified sequences within the BIN, for example, by integrating an R protocol with BOLD. First, the protocol could check for sequences identical to the new one, as full concordance would prevent monophyly among the records previously assigned to the BIN. If no identical sequence exists, an automated Bayesian phylogenetic analysis should be initiated using morphologically identified records. The required computation capacity could be limited by restricting the number of examined sequences, whereby the new sequence and the most similar sequences with the older species assignment must be part of the analyzed sequence group. Necessary R packages for the alignment [32], fasta editing [33], best-fit substitution model detection [34], Bayesian phylogenetic tree reconstruction [35], and tree annotation [36] are available. If the protocol groups all sequences with the old species assignment into a monophyletic branch sister to the newly submitted sequence, supported by a high posterior probability, this may support a BIN split. If old species assignments are non-monophyletic, we recommend sending an automated message with a checklist to the data manager of the new conflicting record within the BIN, asking for possible reasons for misidentification. If the issue persists, we recommend automatic deletion of the species assignment relying on COI-based identifications in the concerned BIN (Table 3).

The detection of records incorrectly assigned to a genus poses a significant post-submission challenge. Such records may be detectable through reduced Bayesian analyses if BINs contain sequences assigned to different genera. Otherwise, detection remains challenging, although whole-genus Bayesian analyses previously enabled clear identification by clustering sequences outside the genus [9]. However, this demands computational resources beyond BOLD servers'

capacity for all submissions. Simpler methods like neighbor-joining approaches are less reliable, as they lack node probability measures [37].

### Enforcement of OTU pipeline criteria and submission requirements

Furthermore, we found records assigned to BINs conflicting with the sequence quality criteria of the BOLD OTU pipeline [1] and missing sample locations in BOLD annotations, standing in contradiction to sequence submission requirements [23]. We encourage investigating and resolving the origin of this type of database error and enforcing rules for submission and sequence BIN assignment (Table 3).

## Conclusion

Despite the aforementioned problems in BOLD v4, we consider BOLD as the most important DNA barcoding reference database due to the large number of COI-5P-based records, a high coverage of metazoan species and a high reliability compared to other COI-5P libraries like GenBank [38]. However, there is a need to improve the BOLD engine in several aspects to avoid the accumulation of incorrect taxonomic assignments in BINs. This has a particular impact on BOLD query results and may give users the impression that BOLD is less reliable, as long as the discussed origin of the described errors and gaps in metadata is not addressed. In BOLD v5, multiple identified systematic problems related to the employed species determination algorithm, BOLD submission process and post-submission process will remain, as RESL still generates BINs and the submission and post-processes still rely on the BOLD v4 engine. In the case of meta-barcoding studies, where biodiversity indices are calculated and compared based on BINs, the current state of BOLD may be sufficient. However, studies using BOLD as a reference library to identify the species' specific ecological features, such as host or prey preferences, might be related to uncertainties.

## Supporting information

**S1 File.** S1 Table. Updated metadata (4th September 2023) of sequence data. S2 Table. BOLD query results. (7Z)

**S2 File.** R project including metadata update. (7Z)

## Acknowledgments

We are grateful all institutions, project leaders and responsible data managers of BOLD, who enabled this study by providing access to their data in the scope of our former research [9]. We acknowledge the German Barcode of Life project, the Centre of Biodiversity Genomics, the Canadian Collection of Insects, the Royal British Columbia Museum, the Natural History Collections of the University Museum of Bergen, the Norwegian Institute, the University of Oslo, the University of Oulu, the Bavarian State Collection for Zoology, and the Naturalis Biodiversity Center for giving us access to private data or publishing private records and for providing supplemental metadata. Finally, we thank the eckstrakt shared office space for making a workspace available during the finalization of the manuscript.

## Author contributions

**Conceptualization:** Frederik Stein, Oliver Gailing.

**Data curation:** Frederik Stein.

**Methodology:** Frederik Stein.

**Supervision:** Oliver Gailing.

**Visualization:** Frederik Stein.

**Writing – original draft:** Frederik Stein.

**Writing – review & editing:** Oliver Gailing.

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
