## [Decision Letter · Decision Letter 0]

14 Jul 2025

Dear Dr. Stein,

Thank you for submitting your manuscript to PLOS ONE. After careful consideration, we feel that it has merit but does not fully meet PLOS ONE’s publication criteria as it currently stands. Therefore, we invite you to submit a revised version of the manuscript that addresses the points raised during the review process.

We look forward to receiving your revised manuscript.

Kind regards,

Vazrick Nazari, PhD

Academic Editor

PLOS ONE

 [Project: Risk Management for Biotic Damage to Ensure Sustainable Forest Management (RiMa; FKZ 22019814) funded by the Federal Ministry of Food and Agriculture/ FNR]. 

Additional Editor Comments (if provided):

Reviewers' comments:

Reviewer's Responses to Questions

**Comments to the Author**

1. Is the manuscript technically sound, and do the data support the conclusions?

Reviewer #1: Yes

Reviewer #2: Yes

Reviewer #3: Partly

2. Has the statistical analysis been performed appropriately and rigorously?

Reviewer #1: Yes

Reviewer #2: N/A

Reviewer #3: No

3. Have the authors made all data underlying the findings in their manuscript fully available?

Reviewer #1: Yes

Reviewer #2: Yes

Reviewer #3: No

4. Is the manuscript presented in an intelligible fashion and written in standard English?

Reviewer #1: Yes

Reviewer #2: Yes

Reviewer #3: Yes

Reviewer #1: Manuscript: “Identification of BOLD engine deficiencies and suggestions for the improvement based on a curated record set”

Recommendation: Minor revisions.

Dear Editor,

After carefully reading the manuscript, my recommendation is “Minor revision”. My suggestions and corrections were made directly in the manuscript pdf file.

In my opinion, this manuscript is extremely relevant, especially for the fields of molecular identification and integrative taxonomy, and deserves to be published. It highlights shortcomings in the BOLD system; however, it also suggests improvements for this important database.

This review is not anonymous, and authors can feel free to contact me.

All the best,

Felipe Ottoni

Reviewer #2: This study critically evaluates the performance of the BOLD identification engine using a curated dataset from the genus Tachina (Tachinidae). It points out major problems in the BOLD system, especially with conflicting species classifications within BINs, weaknesses of the RESL algorithm, and problems with metadata and taxonomic accuracy. By comparing RESL with ASAP, the authors show that ASAP is better at defining species because it can adapt beyond just the barcoding gap idea. The findings emphasize the value of improved submission protocols and stricter post-submission curation.

This article is well written and very detailed, with pertinent data for the analyses. I believe that only concising the discussion section of the article is necessary, as its detailed version is provided in Table 3. Also, can Tachinidae be included in the title, as the work is done on this dipteran group. i have included all the changes in the manuscript in track change mode. Finally, given the rigorous methodology, clear identification of systemic issues, and constructive recommendations for improving the BOLD platform’s reliability, this work is well-founded. I congratulate the authors for such a good work.

Reviewer #3: I read the manuscript with great interest. I support the authors' call for a better discussion of the strengths and weaknesses of BOLD and the BIN system. Especially today, there is a need for clean processes when dealing with automatic species identification. Reference databases depend on sequence and metadata provided by experienced taxonomists.

Summary

The manuscript investigates deficiencies in the BOLD identification engine, focusing on how species assignments within Barcode Index Numbers (BINs) can be inconsistent due to algorithmic and metadata issues. Using curated and non curated BOLD records (812 sequences public and private) from the genus Tachina, the authors attempt to compare the performance of the RESL and ASAP species delimitation algorithms and analyze how query results are affected by different sources of error in the BOLD system. They propose improvements to the submission and post-submission processes to enhance the reliability of species identification.

Strengths

The manuscript addresses some important issues that undoubtedly need more publication and discussion in the community. The use of the Barcode of Life Database (BOLD) for species identification has become increasingly important in recent years and has found its way into taxonomic ecological and conservation biology research. The manuscript takes as its starting point the rapid identification of pests or other ecologically important species. The authors thus address an important aspect of good identification using barcoding from metabarcoding. In particular, the aspects that the database does not cover all species and that there are gaps in the information on the type of identification methods (e.g. morphology) that are used before uploading are a point that is sometimes not considered enough

Major Concerns

The authors attempt to make statements about the quality and methods used in relation to the BOLD data. To do so, it is necessary to select the data in such a way that the results allow conclusions to be drawn.

I see major weaknesses in the size of the data set used as the starting point for the study. Although it may make sense to look at a small part of the data set for certain aspects, it is necessary to look at the entire data set in order to make general statements, since the creation of BINs, clustering, and polishing are based on more than just a subset of the data.

Another important point is that the approach taken in the manuscript raises expectations that the technical problems of query search or clustering will be addressed, rather than simply taking the results of this small data set as a starting point.

The authors attempt to compare Bayesian phylogenetic analyses that use evolutionary models with clustering and genetic distance, which are required for the search. Since the barcoding marker COI cannot be used for deeper phylogenetic analyses, I am critical of using terms such as monophyly or paraphyly to question the BIN assignments.

Lines 344–363: Barcoding gap “values” are placed in a comparative context and discussed. The aspects of population and sample size, geographical distances, and others are mentioned, but the reference to the message of the manuscript is not really clear. It seems (not only there) as if the results are examined in relation to the species and not to the BOLD database. Similar things happen in many places in the text, with the authors jumping back and forth between different levels.

It is striking that there is constant jumping between the results of BIN clustering, searches, and the display on the BOLD GUI (e.g. lines 80 - 87). This makes it difficult to understand the text, and the question posed in the manuscript is not kept clear in mind.

Points 1-5 are just a few examples of issues in this paper that are important for deciding on the status of the manuscript. However, there are other similar issues that create inconsistencies in the narrative and make it hard to follow the line of argument.

Minor Concerns

The minor problems in the manuscript that need improvement are errors in wording that make some sentences difficult to understand (e.g., lines 48, 49).

The manuscript would benefit from a clearer structure, with non-essential details moved to the supplementary methods (e.g., lines 94–97).

Suggestions for Improvement

In its current state, I would recommend a major revision of this paper. Since I generally think this kind of work is a good idea and the authors are pursuing a meaningful goal. I recommend that the authors focus on a more specific topic, such as incorrect data, missing data, or algorithmic shortcomings, and pursue this in a more targeted way.

Recommendation:

Major Revision

**Do you want your identity to be public for this peer review?** For information about this choice, including consent withdrawal, please see our Privacy Policy

Reviewer #1: **Yes: ** Felipe Polivanov Ottoni

Reviewer #2: No

Reviewer #3: No

---

## [Author Response · Author response to Decision Letter 1]

5 Aug 2025

Response to Reviewers

Reviewer 1:

Dear Editor,

After carefully reading the manuscript, my recommendation is “Minor revision”. My suggestions and corrections were made directly in the manuscript pdf file.

In my opinion, this manuscript is extremely relevant, especially for the fields of molecular identification and integrative taxonomy, and deserves to be published. It highlights shortcomings in the BOLD system; however, it also suggests improvements for this important database.

This review is not anonymous, and authors can feel free to contact me.

All the best,

Felipe Ottoni

Authors response:

Dear Prof. Ottoni,

Thank you very much for your positive and encouraging feedback. We greatly appreciate your careful reading of the manuscript and your support for its relevance to molecular identification and integrative taxonomy.

We also thank you for checking the taxonomic authorities and for your helpful corrections made directly in the PDF.

Please find our detailed responses to your specific comments below.

In addition, we made a few minor adjustments elsewhere in the manuscript—such as correcting the number of examined records from 812 to 808 and slightly adjusting some values in Figure 3. We also relabeled the questionable BIN category from “paraphyletic structure” to “non-monophyletic structure,” as this issue involved not only paraphyletic but also polyphyletic cases, including records assigned to the polyphyletic BIN BOLD:AAN9512. These revisions are minor and do not affect the overall results or conclusions of the study.

Best regards,

Frederik Stein and Oliver Gailing

Specific comments:

Reviewer 1:

line 35: Keywords should be presented in a logical order. I suggest alphabetical order. Furthermore, it is not recommended to repeat words in the Keywords section that are already in the title (e.g. BOLD), since this reduces the search power of your work. Please, you should replace the keyword "BOLD". "Molecular identification" might be a more suitable keyword.

Authors response:

Thank you for pointing this out and for your helpful suggestion. We have revised the keywords to follow alphabetical order and replaced “BOLD” with “Molecular identification” to avoid redundancy with the title and to improve searchability, as recommended.

Reviewer 1:

line 37: This is the first time the term appears after the abstract. I suggest providing the full name followed by the abbreviation in parentheses.

line 48:This is the first time the term appears after the abstract. I suggest providing the full name followed by the abbreviation in parentheses.

Authors response:

We have now introduced the full terms followed by their abbreviations upon first mention in the main text after the abstract.

Reviewer 1:

line 68: Tachina Meigen, 1803 (Diptera, Tachinidae); The author and year of genera are never presented in parentheses. Authors and years are presented in parentheses only for species that have changed genus since their original description. Authors and years are presented in parentheses when the species have changed genus since their original description.

line 133: Tachina nitida (Wulp, 1882)

line 196: Tachina ursina (Meigen, 1824); Authors and years are presented in parentheses when the species have changed genus since their original description.

line 208: Tachina ferox Meigen & Panzer, 1806

line 211: Tachina algens Wiedemann, 1830

line 219: Eudoromyia Bezzi, 1906

line 220: Tachina fera (Linnaeus, 1761)

line 234-235: remove first descriptor

line 257: Tachina nupta (Rondani, 1859)

line 307: Nowickia Wachtl, 1894

Authors response:

Thank you for these detailed corrections. We have carefully revised the manuscript to implement all recommended changes to author citations for genera and species, ensuring consistency with accepted taxonomic conventions.

Reviewer 1 other comments on nomenclature :

line 227: What does it mean?

line 259: What does it mean?

line 266: What does it mean?

Authors response:

We have clarified this in the manuscript (lines 108-110) as follows: “One of these OTU names, Tachina nr. magnicornis, refers to a group of individuals considered to represent one morphologically distinguishable taxon, with the abbreviation “nr.” indicating uncertain but close taxonomic affinity [15].”

Reviewer 1:

line 232: Tachina marklini Zetterstedt, 1838

Authors response:

Thank you for the comment regarding Tachina marklini Zetterstedt, 1838. This represents a cryptic species complex, with evidence supporting the existence of both a Palearctic Tachina marklini and a Nearctic Tachina marklini (see Fig. 1). We discuss this in detail in:

Stein F, Gailing O, Moura CCM. Curating BOLD records via Bayesian phylogenetic assignments enables harmonization of regional subgeneric classifications and cryptic species detection within the genus Tachina (Diptera: Tachinidae). Ann Entomol Soc Am. 2024:245-256. doi: 10.1093/aesa/saae01

In the current manuscript, we focus on species-level identities based on the COI locus. The presence of this cryptic species was confirmed through congruent results from RESL, ASAP, and Bayesian phylogenetic analyses, which we used to compare the performance of species delimitation algorithms.

The species designation “Tachina marklini crypt 1” reflects this cryptic diversity detected in our analyses.

Reviewer 1:

line 269: Tachina casta (Rondani, 1859)

Authors response:

Thank you for noting this. The authority for Tachina casta (Rondani, 1859) has already been introduced earlier in the manuscript (line 229), so we did not repeat it here for brevity and consistency.

Reviewer 2:

This study critically evaluates the performance of the BOLD identification engine using a curated dataset from the genus Tachina (Tachinidae). It points out major problems in the BOLD system, especially with conflicting species classifications within BINs, weaknesses of the RESL algorithm, and problems with metadata and taxonomic accuracy. By comparing RESL with ASAP, the authors show that ASAP is better at defining species because it can adapt beyond just the barcoding gap idea. The findings emphasize the value of improved submission protocols and stricter post-submission curation.

This article is well written and very detailed, with pertinent data for the analyses. I believe that only concising the discussion section of the article is necessary, as its detailed version is provided in Table 3. Also, can Tachinidae be included in the title, as the work is done on this dipteran group. i have included all the changes in the manuscript in track change mode. Finally, given the rigorous methodology, clear identification of systemic issues, and constructive recommendations for improving the BOLD platform’s reliability, this work is well-founded. I congratulate the authors for such a good work.

Authors response:

Dear Reviewer,

Thank you very much for your positive and encouraging evaluation of our manuscript. We sincerely appreciate your thoughtful comments and your recognition of the rigor and clarity of our work. We agree with your recommendation to shorten the discussion section. Accordingly, we have revised it to be more concise while preserving the key insights and conclusions of the study.

Thank you also for your suggestion regarding the title. We understand the rationale for including Tachinidae. However, since our analysis was conducted specifically at the genus level (Tachina), not across the entire family, we aimed to maintain taxonomic precision while providing sufficient context. Therefore, we have revised the title to:

“Identification of BOLD engine deficiencies and suggestions for improvement based on a curated Tachina (Diptera) record set.”

We believe this revised title more accurately reflects the scope of our study. Please find below our responses to your specific comments.

Please find our detailed responses to your specific comments below.

In addition, we made a few minor adjustments elsewhere in the manuscript—such as correcting the number of examined records from 812 to 808 and slightly adjusting some values in Figure 3. We also relabeled the questionable BIN category from “paraphyletic structure” to “non-monophyletic structure,” as this issue involved not only paraphyletic but also polyphyletic cases, including records assigned to the polyphyletic BIN BOLD:AAN9512. These revisions are minor and do not affect the overall results or conclusions of the study.

Best regards,

Frederik Stein and Oliver Gailing

Specific comments:

Reviewer 2: Lines 386- 388. This line is confusing. Are you suggesting that RESL is better than ASAP? You have already proven that ASAP is better species delimitation tool than that of RESL.

Authors response:

We have revised the sentence (lines 382–384) for clarity:

“Given that ASAP explicitly incorporates the barcoding gap along with additional parameters, we expect it to perform at least as effectively as RESL in arthropod groups with distinct barcoding gaps.”

Reviewer 2: Lines 393- 394. Please provide the correct reference and also make it in PloS One format of just the number. And the reference is of 2022, the title might confuse as it directly cites another publication in the title.

Authors response.

Thanks for pointing that out. We have revised the citation style accordingly; however, please note that the publication year is correct as stated:

Meier R, Blaimer BB, Buenaventura E, Hartop E, Rintelen T von, Srivathsan A, et al. A re-analysis of the data in Sharkey et al.'s (2021) minimalist revision reveals that BINs do not deserve names, but BOLD Systems needs a stronger commitment to open science. Cladistics. 2022; 38:264–75. doi: 10.1111/cla.12489.

The revised sentence (lines 387–389) now reads:

“While employing additional clustering algorithms may be beneficial [10], this would also increase computational demands and could exceed the performance capacity of BOLD’s servers.”

Reviewer 2: This line is confusing. Rewrite it properly.

Authors response:

Thank you for your comment. We have rewritten the sentence (lines 420–422) as follows:

“Moreover, we recommend that the neighbour-joining tree function annotate each BIN and include sequences from all species within the same genus or subgenus as the matched BIN.”

Reviewer 3:

I read the manuscript with great interest. I support the authors' call for a better discussion of the strengths and weaknesses of BOLD and the BIN system. Especially today, there is a need for clean processes when dealing with automatic species identification. Reference databases depend on sequence and metadata provided by experienced taxonomists.

Summary

The manuscript investigates deficiencies in the BOLD identification engine, focusing on how species assignments within Barcode Index Numbers (BINs) can be inconsistent due to algorithmic and metadata issues. Using curated and non curated BOLD records (812 sequences public and private) from the genus Tachina, the authors attempt to compare the performance of the RESL and ASAP species delimitation algorithms and analyze how query results are affected by different sources of error in the BOLD system. They propose improvements to the submission and post-submission processes to enhance the reliability of species identification.

Strengths

The manuscript addresses some important issues that undoubtedly need more publication and discussion in the community. The use of the Barcode of Life Database (BOLD) for species identification has become increasingly important in recent years and has found its way into taxonomic ecological and conservation biology research. The manuscript takes as its starting point the rapid identification of pests or other ecologically important species. The authors thus address an important aspect of good identification using barcoding from metabarcoding. In particular, the aspects that the database does not cover all species and that there are gaps in the information on the type of identification methods (e.g. morphology) that are used before uploading are a point that is sometimes not considered enough.

Authors response:

Dear Reviewer,

We sincerely thank you for your thoughtful and constructive feedback, particularly regarding the need for a clearer discussion of the strengths and weaknesses of the BOLD and BIN systems. Your comments helped us sharpen the focus of our manuscript and ensure that our message is more accessible to readers across disciplines.

In response to your suggestion to improve the coherence and narrative focus of the discussion section, we carefully revised and condensed the discussion to enhance clarity. While we have preserved the overall structure based on positive feedback from Reviewer 1 and Reviewer 2, we aimed to incorporate your concerns by refining the flow of ideas and reducing redundancy.

Our goal in this revision was to strike a balance between your valuable suggestions and the aspects of the discussion that were appreciated by the other reviewers. We hope that the revised discussion achieves greater thematic consistency and presents our findings and proposals more clearly.

Please find our detailed responses to your specific comments below.

In addition, we made a few minor adjustments elsewhere in the manuscript—such as correcting the number of examined records from 812 to 808 and slightly adjusting some values in Figure 3. We also relabeled the questionable BIN category from “paraphyletic structure” to “non-monophyletic structure,” as this issue involved not only paraphyletic but also polyphyletic cases, including records assigned to the polyphyletic BIN BOLD:AAN9512. These revisions are minor and do not affect the overall results or conclusions of the study.

Best regards,

Frederik Stein and Oliver Gailing

Specific comments:

Major Concerns:

Reviewer 3:

The authors attempt to make statements about the quality and methods used in relation to the BOLD data. To do so, it is necessary to select the data in such a way that the results allow conclusions to be drawn.

1. Reviewer 3:

I see major weaknesses in the size of the data set used as the starting point for the study. Although it may make sense to look at a small part of the data set for certain aspects, it is necessary to look at the entire data set in order to make general statements, since the creation of BINs, clustering, and polishing are based on more than just a subset of the data.

Authors response:

We appreciate your concern regarding the dataset size and its implications for making broader conclusions about BOLD and BIN creation. Our intention was not to generalize across all taxa or the entire BOLD database, but rather to use a case study (the genus Tachina) to highlight how specific errors and inconsistencies can arise within the system. This now specified in the introduction to discussion (lines 342-246).

By focusing on a well-curated, medium-sized dataset, we were able to closely examine the underlying causes of algorithmic performance differences and metadata inconsistencies. While we agree that a more comprehensive analysis across multiple taxa would further strengthen general conclusions, we believe our focused approach is appropriate for demonstrating the types of issues that can occur, and for proposing targeted improvements.

We have clarified this scope in the manuscript to better reflect the limits and intent of our analysis.

2. Reviewer 3:

Another important point is that the approach taken in the manuscript raises expectations that the technical problems of query search or clustering will be addressed, rather than simply taking the results of this small data set as a starting point.

Authors response:

Thank you for this observation. We agree that our manuscript does not provide a full technical resolution to the issues in query search or clustering, but rather uses the analysis of a focused dataset to highlight recurring, systemic problems and to propose potential solutions. Although the specific examples are drawn from a limited set of records, the types of issues we identify—such as metadata inconsistencies and algorithmic misassignments—are rooted in the structure and processes of the BOLD system itself. Therefore, while the frequency of errors is not quantitatively representative, the mechanisms

---

## [Decision Letter · Decision Letter 1]

13 Aug 2025

Identification of BOLD engine deficiencies and suggestions for improvement based on a curated Tachina (Diptera) record set

PONE-D-25-31493R1

Dear Dr. Stein,

We’re pleased to inform you that your manuscript has been judged scientifically suitable for publication and will be formally accepted for publication once it meets all outstanding technical requirements.

Kind regards,

Vazrick Nazari, PhD

Academic Editor

PLOS ONE

Reviewers' comments:

Reviewer's Responses to Questions

**Comments to the Author**

Reviewer #1: All comments have been addressed

2. Is the manuscript technically sound, and do the data support the conclusions?

Reviewer #1: Yes

3. Has the statistical analysis been performed appropriately and rigorously?

Reviewer #1: Yes

4. Have the authors made all data underlying the findings in their manuscript fully available?

Reviewer #1: Yes

5. Is the manuscript presented in an intelligible fashion and written in standard English?

Reviewer #1: Yes

Reviewer #1: Dear editor,

As I mentioned in my review of the previous version of the manuscript, this is an extremely relevant paper, especially for the fields of molecular identification and integrative taxonomy, and deserves to be published. It highlights shortcomings in the BOLD system; however, it also suggests improvements for this important database.

After carefully reading the revised version of the manuscript and the rebuttal letter sent by the reviewers, I I noticed that all of my corrections and suggestions have been properly addressed. Therefore, I recommend the publication of the manuscript in its current form.

This review is not anonymous.

All the best,

Felipe Ottoni

**Do you want your identity to be public for this peer review?** For information about this choice, including consent withdrawal, please see our Privacy Policy

Reviewer #1: **Yes: ** Felipe Polivanov Ottoni

---

## [Editor Report · Acceptance letter]

PONE-D-25-31493R1

PLOS ONE

Dear Dr. Stein,

I'm pleased to inform you that your manuscript has been deemed suitable for publication in PLOS ONE. Congratulations! Your manuscript is now being handed over to our production team.

Kind regards,

on behalf of

Dr. Vazrick Nazari

Academic Editor

PLOS ONE